# Role of ctDNA in Breast Cancer

**DOI:** 10.3390/cancers14020310

**Published:** 2022-01-09

**Authors:** Marta Sant, Adrià Bernat-Peguera, Eudald Felip, Mireia Margelí

**Affiliations:** 1Medical Oncology Department, Catalan Institute of Oncology-Badalona, Hospital Germans Trias i Pujol (HGTiP), 08916 Badalona, Spain; 2Badalona Applied Research Group in Oncology (B-ARGO), Institut d’Investigació en Ciències de la Salut Germans Trias i Pujol (IGTP), Universitat Autònoma de Barcelona, 08916 Badalona, Spain

**Keywords:** ctDNA, breast cancer, personalized medicine, liquid biopsy, cancer diagnosis

## Abstract

**Simple Summary:**

Circulating tumor DNA is DNA released by the tumor into the bloodstream. In breast cancer, it is used mainly in research or in clinical trials, but it will likely be used in routine clinical practice once certain issues have been worked out and methods of analysis have been improved and standardized. Breast cancer classification and treatment selection are now based on analysis of the tumor but circulating tumor DNA carries many features of the original tumor and can be analyzed from a simple, non-invasive blood extraction. Here, we review its potential role in early breast cancer (for screening, diagnosis, detection of minimal residual disease after surgery, follow up, and treatment) and in metastatic breast cancer (for the detection of mutations, prognosis and treatment).

**Abstract:**

Breast cancer is currently classified by immunohistochemistry. However, technological advances in the detection of circulating tumor DNA (ctDNA) have made new options available for diagnosis, classification, biological knowledge, and treatment selection. Breast cancer is a heterogeneous disease and ctDNA can accurately reflect this heterogeneity, allowing us to detect, monitor, and understand the evolution of the disease. Breast cancer patients have higher levels of circulating DNA than healthy subjects, and ctDNA can be used for different objectives at different timepoints of the disease, ranging from screening and early detection to monitoring for resistance mutations in advanced disease. In early breast cancer, ctDNA clearance has been associated with higher rates of complete pathological response after neoadjuvant treatment and with fewer recurrences after radical treatments. In metastatic disease, ctDNA can help select the optimal sequencing of treatments. In the future, thanks to new bioinformatics tools, the use of ctDNA in breast cancer will become more frequent, enhancing our knowledge of the biology of tumors. Moreover, deep learning algorithms may also be able to predict breast cancer evolution or treatment sensitivity. In the coming years, continued research and the improvement of liquid biopsy techniques will be key to the implementation of ctDNA analysis in routine clinical practice.

## 1. Introduction

### 1.1. The Role of Circulating Tumor DNA (ctDNA) in Breast Cancer

Despite improvements in breast cancer detection, prevention, and treatment, there are more than 2.3 million new cases worldwide, causing more than 650,000 estimated deaths per year [1,2]. Regardless of the exact origin of malignant cells, breast cancer is a heterogeneous disease. The primary classification of breast cancer is based on immunohistochemistry markers in tumor biopsies: estrogen receptor (ER), progesterone receptor (PR), KI-67, and human epidermal growth factor 2 (HER2). Different subtypes have been proposed with the aim of personalizing treatment and prognosis [3,4], and some groups have suggested using gene expression profiles to characterize five different intrinsic molecular subtypes of breast cancer, with different outcomes (luminal A, luminal B, HER-2 enriched, basal-like, and claudin-low) [5,6,7]. 

At present, the selection of breast cancer treatment is based on the analysis of tumor biopsy. However, the information obtained from the tumor biopsy is not permanent, and changes and acquired resistance that can occur during cancer treatment cannot be evaluated or analyzed in the original tumor specimen. Studies have found up to 25% of changes in subtypes at or after progression to anticancer therapies [8,9]. Although tumor biopsy is still the gold standard for diagnosis, classification, and treatment decisions, there is a growing interest in improving precision medicine by characterizing and monitoring the tumor genome in blood samples [10], known as liquid biopsy. A liquid biopsy can contain circulating tumor cells (CTCs), ctDNA, and exosomes that can help to understand tumor evolution, resistance, and heterogeneity during treatment [11]. Furthermore, in some cases, a tumor biopsy may not be feasible, and a liquid biopsy would be the only method to obtain a diagnosis or knowledge of the tumor biology. 

### 1.2. ctDNA 

In the late 1940s, extracellular nucleic acid was observed in human plasma from a patient with a systemic lupus erythematosus [12]. Since then, technological developments have allowed ctDNA to be isolated from cancer patients’ blood samples. It is known that cancer patients have higher levels of circulating DNA than healthy subjects, and this free DNA can be used for different purposes [13]. In healthy donors, circulating free DNA (cfDNA) is isolated primarily from hematopoietic cells [14]. However, the origin of ctDNA is more complex. Cancer produces not only local infiltration but also malignant cells that are released into the lymphatic or vascular system. CTCs in the blood could be responsible for metastatic progression. The origin of ctDNA is thought to be the cellular breakdown from the tumor through apoptosis, necrosis or phagocytosis, although it could also arise from CTCs and active secretion from cellular structures has also been described [15,16]. 

To understand how to analyze ctDNA, we must first know its main characteristics. Circulating DNA comprises short fragments of DNA, most of which are around 180 bp, and ctDNA is not an exception. The half-life of ctDNA is short. It has been estimated that in colorectal cancer patients who have undergone complete resection, the half-life of ctDNA is less than two hours [17]. However, recent studies on cfDNA collection and processing have found that cfDNA levels are stable for 24 h at room temperature or even for 3 days stored at 4 °C using EDTA tubes [18]. These factors need to be taken into account if ctDNA detection is to be progressively implemented in routine clinical practice in most hospitals. 

Another important consideration is the ability to quantify ctDNA, and the variant allele fraction (VAF) is a crucial parameter. VAF is the percentage of sequence reads detected fitting specific DNA by complete coverage at the locus. Therefore, VAF could be the proportion of DNA carrying the mutant variant [19]. VAF detection in cancer patients can vary; for example, more than 10% could be detected in the metastatic setting, while in early stages of cancer or in minimal residual disease (MRD), less than 0.1% might be detected [20].

The applications of ctDNA are moving fast. Since the FDA approved the first blood test in 2016, liquid biopsy has started to be considered as the new standard for detecting EGFR mutations in lung cancer and for selecting select targeted therapy [21], and several investigators have used cfDNA to monitor metastatic cancer. Moreover, the mean of all VAF ratios was shown to predict progression-free survival in metastatic breast cancer patients under treatment with endocrine therapy plus cyclin-dependent kinase (CDK) inhibitors [22]. 

## 2. Methodology

The PubMed database was searched from July 2014 to July 2021 for ctDNA in breast cancer. Selected key search words were breast cancer, early breast cancer, metastatic breast cancer, ctDNA, circulating free DNA, cancer heterogeneity, breast cancer diagnosis and precision oncology. The authors screened and selected most relevant articles to review.

Parts of Figure 1 were drawn by using pictures from Servier Medical Art. Servier Medical Art and the Figure modified with text, markings (stars), and annotations after the adaptation of “ctDNA” from Servier Medical Art by Servier, licensed under a Creative Commons Attribution 3.0 Unported Licens..

## 3. Methods for ctDNA Detection and Analysis

In cancer patients, ctDNA is found in a variable but usually very low percentage (0.01–1.0%) of the total cfDNA, which is usually less than 1 ng/μL, and varies depending on the stage, location, or vascularization of the tumor [23]. The amount of ctDNA is known to be 2–24 times higher in serum than in plasma [24]. However, this higher amount is associated with contamination by DNA released by blood cells during the coagulation process, so the use of plasma for ctDNA analysis is recommended [25]. It is clear that high-throughput isolation processes and highly sensitive detection methods are needed to detect, monitor, and characterize this ctDNA. The method of cfDNA extraction and quantification is crucial to achieve high isolation yields and provide information about the cfDNA molecule size distribution, which conditions detection sensitivity. 

In the last 5 years, we have witnessed a true technological revolution that has made it possible to achieve sufficient sensitivity to study different genomic alterations, such as point mutations, small indels, gene rearrangements, chromosomal gains or losses, and epigenetic alterations, in ctDNA from patients with different types of cancer. 

qPCR and Sanger sequencing used to be very useful techniques, but due to their low sensitivity, they have been superseded by others. Targeted techniques have been developed, such as droplet digital polymerase chain reaction (ddPCR) and beads, emulsion, amplification and magnetics (BEAMing). ddPCR is based on a first emulsion step, in which the formation of thousands of lipid droplets allows the individualization of DNA fragments, and a second step where PCR is carried out inside each droplet from a single ctDNA molecule. Mutated and unmutated copies are detected by using specific primers labelled with different fluorophores. In a prospective study published by Beaver et al. [26], DNA from 30 breast cancer tumors and paired plasma samples before and after surgery was analyzed for PIK3CA mutations by ddPCR. Of the 15 PIK3CA mutations detected in tumors, 14 corresponding mutations were detected in ctDNA before surgery, and in half of the patients after surgery, demonstrating that liquid biopsy can give precise information on MRD. Although it can only be used to screen for known mutations and specific methylation sites, ddPCR allows the absolute quantification of the initial sample with a high sensitivity (0.01–0.1%) [27]. 

BEAMing is based on a first step of specific pre-amplification of interest regions followed by an emulsion PCR using magnetic particles coated with specific primers in which amplicons bind and are again amplified within a lipid droplet. Then the oil droplets are fragmented and all magnetic particles, with amplified regions attached, are hybridized with probes labelled with differential fluorophores for the mutated and non-mutated sequence, allowing discrimination of both fractions by flow cytometry. Like ddPCR, BEAMing is highly sensitive (0.01%) and can only screen for known mutations and specific methylation sites. Due to the complex workflow of this technique, its possible implementation in routine clinical practice seems difficult [27]. 

Other targeted DNA sequencing techniques include tagged amplicon deep sequencing (TAM-Seq), cancer personalized profiling by deep sequencing (CAPP-Seq), safe sequencing system (Safe-Seq), and amplicon sequencing (AmpliSeq). These techniques are very useful for analysis of a limited panel of potential mutations in the primary tumor or biopsy specimens [23,27]. For example, the Oncomine Breast cfDNA (Thermofisher, Waltham, MA, USA) test, based on AmpliSeq technology, is being used in clinical practice to detect aberrations in a limited number of genes in samples from breast cancer patients. New techniques are also being developed, such as targeted digital sequencing (TARDIS), which employs simultaneous deep sequencing of patient-specific somatic mutations to improve analytical and quantitative precision for ctDNA analysis [28].

Nevertheless, as these targeted techniques can only analyze a limited number of mutations at a time, the tumor heterogeneity represented in the ctDNA can be lost. Moreover, prior individual mutational information from the tumor is required. To resolve these issues, massive next-generation sequencing (NGS) techniques have been developed. These panels are based on a genome-wide analysis of copy number aberrations (CNAs), point mutations, and other genetic aberrations by whole-genome sequencing (WGS) or whole-exome sequencing. This approach can be exploited for monitoring mutations during treatment, for de novo discovery of genetic changes underlying therapy resistance, for identifying new actionable targets, and for characterizing mutational loads to guide potential immunotherapy. However, the lower overall sensitivity (1–5%) and the need for higher concentrations of ctDNA are drawbacks that limit the utility of these techniques in patients with low ctDNA load. 

Finally, and importantly, a large proportion of cfDNA comes from blood cells, and some of the somatic variants identified by NGS are known to be a consequence of clonal hematopoiesis. This issue is still not completely clarified, and it is recommended that the blood cell fraction be included in sequencing studies together with the ctDNA in order to avoid false positives [29]. 

### Challenges for ctDNA in Breast Cancer

The implementation of ctDNA analysis in routine clinical practice faces several challenges. For example, it is crucial to improve detection of the low fraction of ctDNA in cfDNA and to identify tumor mutations in plasma at VAFs below the background sequencing error threshold. New detection methods have recently been described that can overcome this hurdle [30]. Blood volume is another important issue. To detect a single mutation with a VAF of 0.01% with 95% confidence requires 150–300 mL with 30,000× sequencing coverage, but increasing the numbers of mutations detected would also increase the volume needed [31]. In summary, the sensitivity of ctDNA analysis in localized breast cancer depends on both the blood volume analyzed and the number of mutations screened.

## 4. ctDNA in Early Breast Cancer

### 4.1. ctDNA in Breast Cancer Diagnosis

At present, the only available methods for screening and early detection of localized breast cancer, for which there is the option of curative treatment, are self-exploration and imaging tests such as mammography, echography, and magnetic resonance imaging. Mammography is a sensitive test that is recommended every two years in women older than 50. However, about 20% of patients diagnosed with breast cancer are younger than 50 and there is no specific test for this age group [32]. Moreover, screening by mammography has been related to overdiagnosis [33]. To resolve these issues, it is clear that new methods are needed, and liquid biopsy is a promising option. For instance, cfDNA levels were higher in early breast cancer patients compared to those with benign breast lesions, and cfDNA levels decreased after surgery. In addition, ctDNA levels correlated with tumor size and nodal involvement [34].

Around 80–85% of breast cancers are diagnosed at the early stage but, unfortunately, about 30% of these will relapse with metastatic progression during the follow-up period. Primary risk factors for relapse are well described, including tumor grade, tumor stage, lymph node involvement, and immunohistology characteristics, and are used to define the best therapeutic approach [35]. The gold standard for diagnosis is still tissue biopsy, which provides information on the histology, molecular biology, and genetic profile of the tumor. However, breast cancer is a heterogeneous disease, and several molecular alterations may occur over time and influence treatment response, making it necessary to monitor these modifications and personalize treatment accordingly. The use of liquid biopsy is a promising option for improving medical precision in oncology due to its power to detect driver mutations, but low levels of ctDNA in early breast cancer may be challenging. Some initiatives, such as the CancerSEEK blood test, which is designed to combine ctDNA and protein biomarkers, presented a sensitivity of 73% in stage II and 79% in stage III disease but less than 43% in stage I, and a specificity of over 99%. However, before this kind of test can be implemented in routine practice, it needs to be validated in larger prospective studies [36]. 

Other potential approaches to the use of cfDNA and ctDNA in early breast cancer are emerging. Interestingly, global cfDNA can be easily quantified and is known to be increased in breast cancer patients compared to healthy subjects [37]. Moreover, high levels of cfDNA correlate with more advanced disease stages [38]. A more complex approach for breast cancer screening uses multiplexed PCR and NGS to identify both clonal and subclonal copy-number variants (CNVs) in the ctDNA of breast cancer patients [39]. Encouragingly, a meta-analysis has reported a sensitivity of 88% and a specificity of 98% using qualitative ctDNA for screening [40]. However, global ctDNA levels are lower in early than in metastatic breast cancer patients, and the effectiveness of this method as a universal screening or diagnostic tool should be assessed in large prospective trials [41]. Moreover, some approaches based on the detection of driver mutations in cfDNA could be hampered by CHIP-related mutations, which are present in plasma samples from patients without cancer [42]. This problem could be ameliorated through the detection of genomic alterations that are specific to tumor type [43].

### 4.2. ctDNA-Based Follow-Up Assessments

Early diagnosis of relapse after a complete primary breast cancer resection is a priority for oncologists. Routine mammography, clinical exploration, symptoms anamnesis, and routine laboratory analyses are recommended for follow-up visits to detect distant relapse [44]. With the improvement of techniques, the implementation of ctDNA during follow up has been assessed in several studies. For example, a prospective study by Garcia-Murillas et al. sequenced 14 breast cancer driver gene mutations detected in primary tumor biopsies from 55 patients and 45 patients carried at least 1 of these mutations in ctDNA. The persistence of detectable mutations in ctDNA 2–4 weeks after surgery was the most reliable parameter associated with a high risk of early relapse [45]. In addition, Olsson et al. performed a retrospective study of 20 patients, 14 of whom relapsed [46], and Coombes et al. performed a prospective study of 49 patients, 18 of whom relapsed [47]. Both studies included patients with non-metastatic (stage I–III) breast cancer at the start of the study and found that serial monitoring of ctDNA was able to detect metastatic progression on an average of 11 months (range, 0.5–37) before detection by clinical manifestation, imaging, CA 15-3 test, or liver function determination, with a sensitivity of 86–93% and a specificity of 100%. 

For those patients in whom common mutations are not found, patient-specific ctDNA (also known as personalized DNA)—targeting variants selected from the primary tumor exome—can be a good alternative for detecting relapse after primary treatments. However, this method is limited in that it cannot detect a second primary breast cancer [46]. 

### 4.3. Detection of MRD

The principal neoadjuvant and adjuvant chemotherapy regimens in breast cancer include anthracyclines and taxanes, which are associated with short- and long-term toxicities. Detecting the need to increase or reduce the dose or duration of treatment could decrease these toxicities and also improve overall survival [48]. For this reason, several clinical trials use radiological tests during neoadjuvant treatment to predict a pathological complete response (pCR) and personalize treatment duration and dosage based on these findings [49]. However, ctDNA could be a more sensitive method to evaluate treatment response. In addition, ctDNA analysis could help identify patients unlikely to benefit from adjuvant chemotherapy and could play a crucial role in detecting patients with micro-metastases and a higher risk of future distant metastases, thus improving patient selection for certain treatments and avoiding unnecessary adverse events.

Detecting and monitoring MRD could be crucial for evaluating treatment response and guiding subsequent therapeutic decisions. The presence of PIK3CA or TP53 mutations before neoadjuvant therapy was associated with a lower rate of pCR in the NeoALTTO trial, suggesting that a more aggressive or targeted treatment approach could be proposed to patients carrying these mutations [50]. High ctDNA levels prior to neoadjuvant treatment have been associated with tumor size, aggressivity, and subtype. Interestingly, the presence of ctDNA after neoadjuvant treatment has also been associated with lower pCR rates, while the clearance of ctDNA after treatment was associated with longer survival even in patients who did not achieve pCR [51] (Table 1). 

### 4.4. Epigenetic ctDNA Alterations

Since gene methylation and transcriptional regulation could predict treatment response and patient outcome, epigenetic ctDNA alterations have also been proposed as a promising biomarker in early breast cancer [56,57]. Serial blood samples taken during neoadjuvant chemotherapy were analyzed for the methylation status of BRCA1, MGMT, GSTP1, Stratifin, and MDR1. BRCA1 methylation frequency was different in responders and non-responders [58]. Another study of 336 early breast cancer patients found that patients with methylation of GSP1, RASSF1a and RARb2 promotors before surgery had a lower overall survival rate at eight years than those without methylation (78% vs. 95%) [59]. Measurement of ctDNA methylation has also been proposed as a method to predict resistance to adjuvant tamoxifen treatment [60], and serum DNA methylation has been proposed as a surrogate marker of tumor DNA methylation for diagnosis and prognosis [61].

## 5. ctDNA in Metastatic Breast Cancer

In contrast to early, non-metastatic breast cancer, ctDNA is detectable in the majority of metastatic breast cancers. Zhou et al. reported that 85.71% of stage IV/M1 patients carried tumor-derived mutations in blood, compared to only 57.81% of stage I–III/M0 patients [62]. 

The analysis of ctDNA offers a wide range of information in metastatic breast cancer patients. For example, it can provide a prompt diagnosis of disease relapse in previously treated early breast cancer patients. In addition, the assessment of gene mutations in ctDNA can help to select the best therapy for each patient. ctDNA analysis also provides information on the clonal evolution and heterogeneity of the tumor and can be used in the follow up of the disease to detect response or failure to ongoing treatments and determine prognosis. All of this information is crucial for clinical decision-making and patient management [63,64,65,66].

### 5.1. Tumor Burden Dynamics and Response to Treatment

Fluctuations in ctDNA levels correlate with tumor burden, which makes ctDNA an excellent, non-invasive tool for monitoring tumor evolution, predicting treatment response, and determining prognosis, as shown by Dawson et al. in their prospective study of 30 women with metastatic breast cancer [63,67]. ctDNA is more abundant than CTCs but is also more dynamic and is rapidly cleared from circulation within hours. Furthermore, ctDNA in metastatic breast cancer patients has been shown to accurately represent the mutational profile of individual CTCs. Moreover, an increase in ctDNA levels was able to predict disease progression several months before standard imaging techniques and was able to assess treatment response earlier than any other markers [66,67]. However, at present, ctDNA has not yet been validated for use in routine practice. 

### 5.2. Prognostic Markers

ctDNA percentage—the number of mutant molecules over the total number of molecules at a given genomic position—is quantitatively associated with outcome, with increasing levels of ctDNA associated with shorter overall survival. This relationship does not hold true for invariable biomarkers, such as T, N, histological grade, ER, PR, HER2, and the Nottingham prognostic index [46,66,68]. Moreover, the study by Dawson et al. demonstrated that while CA 15-3 levels were not a prognostic factor, PIK3CA and TP53 mutations in ctDNA were an early indicator of response to treatment [67]. In the INSPIRE phase II basket study and the LOTUS randomized phase II trial, ctDNA levels correlated with progression-free survival, overall survival, and overall clinical response rate [69].

### 5.3. Genetic Heterogeneity and Clonal Evolution

ctDNA can also be used to study clonal evolution during treatment and at progression without the need for repeated biopsies, which may not even be feasible if the tumor is in an inaccessible site [63,70]. Due to this difficulty in performing biopsies of metastatic lesions, the phenotype of the primary tumor most often determines treatment decisions in metastatic breast cancer; however, this may lead to inaccurate decisions, since the genetic make-up of the tumor may change over time [71]. Moreover, a biopsy, from either the primary tumor or the metastasis, may not reflect intratumor heterogeneity, as the biopsy specimen may not be representative of all the tumor cells [63,70]. In contrast, ctDNA can provide insight into the genomic make-up and heterogeneity of inaccessible metastatic lesions [70], which is crucial for detecting the emergence of resistant clones and possible new driver mutations. Furthermore, ctDNA can provide information on the current status of the disease, which can help guide clinical management and the choice of the appropriate targeted therapy during follow up [62,70,71,72,73].

### 5.4. ctDNA Quantification and Gene Mutations 

Assessment of the ctDNA percentage can help determine tumor dynamics, treatment response, and risk of relapse. ctDNA percentage correlated with progression-free survival in triple-negative breast cancer patients [74,75]. In addition, it can be used to assess specific gene mutations. Several genes play an important role in the management of patients with metastatic breast cancer, with TP53, PIK3CA, ESR1, GATA3, ARID1A and PTEN are the most frequently altered [76]. These mutations can be truncal, when they are found in all the patient’s cancer cells, or subclonal, when they are randomly dispersed throughout the genome. The ctDNA dynamics of subclonal mutations have a limited potential to predict clinical outcome [77]. 

Different mutational processes often generate different combinations of mutation types, known as “signatures”. Alexandrov et al. analyzed nearly five million mutations from more than 7000 cancers and identified more than 20 distinct mutational signatures, five of which, including signatures 1 and 2, were prevalent in breast cancer. Hormone receptor (HR)-negative/HER2-positive breast cancers are enriched for age-related signature 1, which is characterized by C > T substitutions at NpCpG trinucleotides [76,78]. In contrast, signature 2 is characterized by C > T and C > G base substitutions at TpCpN trinucleotides. The authors suggest that this signature is due to overactivity of APOBEC family members of cytidine deaminases, which convert cytidine to uracil, together with base excision repair and DNA replication activity [78]. In patients with advanced ER-positive/HER2-negative breast cancer, it seems that APOBEC mutagenesis promotes clonal evolution [79]. Mutations can confer resistance or can be targeted by certain treatments. Additionally, the tumor mutational burden gives information on the immunogenicity of the tumor and is a predictive marker of the response to immunotherapy [63,66,68].

### 5.5. ctDNA Gene Alterations in Metastatic Breast Cancer

*PIK3CA* encodes for the p110a subunit of PI3K. PIK3CA mutations are associated with worse prognosis [80], although they confer sensitivity to PI3K inhibitors (PI3Ki) such as taselisib, alpelisib, buparlisib and copanlisib [71,81,82]. The majority of PIK3CA mutations are truncal mutations, including H1047R/L, N345K, G1049R, E545K and E542K, but others are subclonal [76,77]. Although there are no validated predictive biomarkers of response to CDK 4/6 inhibitors, early ctDNA dynamics of PIK3CA truncal mutations predicted sensitivity to palbociclib, a CDK 4/6 inhibitor. Palbociclib is a cytostatic drug, and its effects decrease PIK3CA-mutant ctDNA, indicating that ctDNA PIK3CA mutations may be useful as an early predictor of response, as was observed in the PALOMA-3 trial of ER-positive/HER2-negative advanced breast cancer patients who had previously progressed to endocrine therapy [69,77]. However, Razavi et al. found that in HR-positive metastatic breast cancer, PTEN loss promotes PIK3alf-independent activation of AKT, causing resistance to PI3Ki [83]. 

ESR1 encodes for an ER, and its mutations are found in 30% of patients receiving endocrine therapy. However, if a CDK 4/6 inhibitor is used together with aromatase inhibitors, the ESR1 mutation rate decreases [66]. ESRI mutations are located in the ligand-binding domain and are hormone-independent activating mutations [84,85]. In some cases, methylation of the ESRI promoter causes gene silencing, leading to a lack of ER expression and resistance to endocrine therapy [65,69,85]. Activating ESR1 mutations are acquired mutations and not a clonal selection, as they are not detected in primary breast cancer and they are found in the subclonal population [65,82,83,84,86].

The ongoing PADA-1 phase III trial is assessing patients with metastatic hormone-sensitive breast cancer treated in the first line with palbociclib plus an aromatase inhibitor. ESRI mutations are analyzed in cfDNA at regular intervals, and at the emergence of ESRI mutations, patients are randomized to continue with aromatase inhibitors or to switch to fulvestrant, a competitive ER antagonist. Preliminary results show that early clearance of ESRI mutations during treatment may greatly reduce the risk of recurrence and that ESR1 mutations are twice as prevalent (7% vs. 3%) among patients who had received aromatase inhibitors in the adjuvant setting [87]. 

Retinoblastoma (RB1) mutations can arise following treatment with CDK 4/6 inhibitors. In the PALOMA-3 trial, RB1 mutations were present in 5% of the patients who progressed during treatment with palbociclib plus fulvestrant but not in those treated with a placebo plus fulvestrant. However, these mutations are likely subclonal and of relatively low prevalence, suggesting that they are not a major mechanism of resistance to CDK 4/6 inhibitors, contrary to what had been suggested in a previous study [79]. In fact, survival data showed that the low rate of RB1 mutations present in palbociclib-treated patients had no detectable effect on either overall survival or sensitivity to subsequent therapies after progression [88]. 

Acquired HER2 mutations confer sensitivity to HER2-targeted therapies, such as neratinib, in HER2-negative (non-amplified) metastatic breast cancer [82]. The HER2 L755S mutation confers resistance to lapatinib but sensitivity to neratinib, both of which bind to the HER2-activating kinase domaine [89]. ctDNA sequencing identified HER2 L869R and HER2 D769Y at the baseline. These mutations decreased during treatment with neratinib and increased at progression, when several other HER2 mutations also emerged, including the T798I mutation in the HER2 kinase domain, which is analogous to the EGFR T790M “gatekeeper” resistance mutation [90]. In women with HER2-positive breast cancer, the HER2 V777L mutation induces acquired resistance to trastuzumab [91,92].

Other genes are also frequently altered in metastatic breast cancer, including *TP53*, *GATA3* and *ARIDIA*, while other gene alterations appear less frequently, such as *ERBB2* (*HER2*) (with mutations or amplifications), *CCND1*, *AKT1*, *ATM*, *BRCA1*, *MYC*, *PB1*, *KRAS*, *SMAD4*, and *BRAF*. The druggable target mutations detected by ctDNA in metastatic breast cancer are *PIK3CA*, *ESR1*, *HER2*, *PTEN* and *AKT1* [82] (Table 2). *AKT1* mutations are truncal and respond to capivasertib, an AKT kinase inhibitor, as found by Turner et al. [82] 

## 6. Future Perspectives

Before the potential future clinical applications of ctDNA are implemented, several prerequisites will need to be met: a reproducible and validated technique, a demonstrable clinical utility, and a cost-effective procedure. At the moment, some large assay platforms are being used to perform analyses in central laboratories, which can add useful information to the classic pathological diagnosis performed at each hospital. Centralization of ctDNA analysis makes it possible to perform multigene assays requiring specialized knowledge and greater resources. Alongside the NGS platforms themselves, adequate cloud storage space is necessary for all generated data. In addition, there is a need for technicians for web-based laboratory tasks and bioinformaticians for data analysis, as well as experts in the field of data interpretation. Central laboratories can provide all this technology and offer their services to different hospitals that may lack the resources necessary to perform these analyses on site. 

If oncology researchers are able to consolidate all the available data and share our efforts and our clinical information, we will hopefully be able to re-understand breast cancer and potentially generate new molecular classifications based on the presence of different ctDNA alterations. For example, luminal breast cancer with PIK3CA and TP53 mutations could have a completely different prognosis than luminal breast cancer without these mutations, and consequently, treatment approaches would be different as well. 

Interestingly, artificial intelligence is already being used to predict drug response, mainly in preclinical models [93,94]. If ctDNA analysis becomes more available and if results are linked to clinical features, tumor evolution, and drug response, new algorithms will appear for predicting drug sensitivity and suggesting new drug combinations.

## 7. Conclusions

In conclusion, the application of liquid biopsy and ctDNA analysis in breast cancer opens a window of opportunity that encompasses all possible disease situations: from early diagnosis, through the detection of MRD, the early detection of relapse, and the monitoring and treatment planning for advanced disease.

However, there are numerous challenges that must be addressed, including the improvement of early detection of breast cancer, where the detection of carcinoma in situ, for example, is still uncertain. However, the inclusion of liquid biopsy in current research projects and clinical trials hopefully presages its implementation in clinical practice in the not-too-distant future: for response monitoring, early detection of progression, detection of MRD after neoadjuvant therapy, early detection of hormonal resistance mechanisms, guiding successive treatments, and obtaining information on the tumor when a biopsy is not feasible.

It is necessary to increase technological efforts to improve the techniques for the use of liquid biopsy and to generate standardized protocols for its implementation in the clinic.Although there is still a long way to go, liquid biopsy has great potential as an essential player in the future clinical approach to breast cancer.

## Figures and Tables

**Figure 1 cancers-14-00310-f001:**
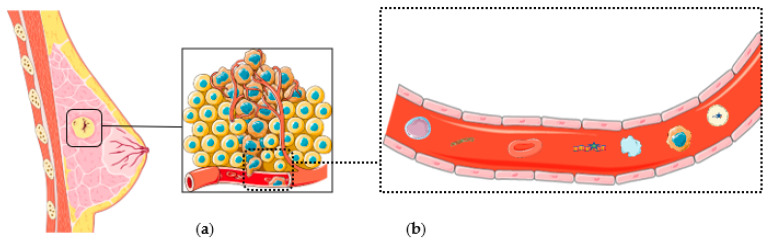
(**a**) Early breast cancer and relation between breast cancer cells and the vascular system. (**b**) Main components found in blood samples, from left to right: lymphocyte cell, cfDNA, erythrocyte, ctDNA, platelet, CTC, exosome with ctDNA.

**Table 1 cancers-14-00310-t001:** Early breast cancer studies monitoring ctDNA in the neoadjuvant chemotherapy setting [28,50,51,52,53,54,55].

Study	Technique	Method	ctDNA/Total SAMPLES	Main Findings
Riva et al. (2017)	ddPCR	Customized panel to track TP53 mutations previously characterized in tumor tissue	38/41	Customized panel detected 75% at baseline; Slow decrease in ctDNA during neoadjuvant chemotherapy was associated with shorter survival
Garcia-Murillas et al. (2019)	ddPCR	Primary tumor was sequenced and personalized tumor-specific ddPCR was used	101/170	ctDNA detection during follow up was associated with a high rate of relapse
Rothé et al. (2019) (NeoAllto trial)	ddPCR	PIK3CA and/or TP53 mutations	69/455	ctDNA detection before neoadjuvant anti-HER2 therapy was associated with low pCR rates
McDonald et al. (2019)	Targeted digital sequencing (TARDIS)	Exome sequencing of tumor biopsies and analysis of dozens to hundreds of mutations in serial plasma samples	33/33	TARDIS results were informative in 100% of the samples; Patients with pCR showed a large decrease in ctDNA concentration during therapy
Radovich et al. (2020)	NGS	Commercial platform covering multiple genes. (FoundationACT^®^ or FoundationOneLiquid Assay^®^)	142/196	Detection of ctDNA and CTCs in triple-negative breast cancer patients after neoadjuvant therapy was associated with disease recurrence
Magbanua et al. (2021)	NGS	Personalized ctDNA test to detect up to 16 patient-specific mutations	61/84	Lack of ctDNA clearance predicted poor response and metastasis
Po-Han Lin et al. (2021)	NGS	Deep sequencing of a target gene panel (14 genes)	60/90	The presence of ctDNA after neoadjuvant therapy was a robust marker for predicting relapse in stage II-to-III breast cancer patients

**Table 2 cancers-14-00310-t002:** Druggable target gene alterations detected in ctDNA in metastatic breast cancer. * (asterisk) means translation termination (stop) codon.

Gene (Hotspot Mutations)	Effect on Treatment Response
PTEN (R130Q, R130G, R130*, R130P, R130Qfs*4)	Sensitivity to capivasertib/ipatasertib (AKT inhibitors) + paclitaxel Resistance to PI3Ki (loss of *PTEN*)
PIK3CA (H1047R, H1047L, N345K, E545K, E542K, E726K)	Resistance to endocrine therapy (truncal mutations) Sensitivity to PI3Ki (taselisib, alpelisib, buparlisib, copanlisib
ESR1 (Y537C, Y537N, Y537S, S463P, D538G)	Resistance to endocrine therapy (subclonal mutations)
AKT (E17K)	Sensitivity to capivasertib (AKT kinase inhibitor)
HER2 (L755S, V777L)	HER2 inhibitor (bind to kinase domain) (lapatinib, neratinib)

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
