# Peer review of "Role of ctDNA in Breast Cancer"

_cancers, 2022, doi:10.3390/cancers14020310_

Round 1

Reviewer 1 Report

The authors reviewed the role of ctDNA in breast cancer including early and metastatic state. Some enhancement are required to improve this article.

1) 2. Methodology: please provide the applied search terms and numbers of found articles. How many articles were included in the screening process?

2)  Table 1: detailed information for sample type and storage condition should be provided in the text because it is important for ctDNA results. 

In addition, detailed platforms for ddPCR and NGS should be provided because they can influence on the ctDNA results.

Further, explanations for TARDIS in text would contribute to better understanding

3)  5.5 ctDNA Gene Alterations in Metastatic Breast Cancer: hot spots for these genes and associated clinical implications should be discussed. For example, the most frequently identified ESR1 mutations are D538G, Y537N, and Y537S.

Author Response

Methodology and table 1 have been revised. The sample collection is not well described in all trials. In most cases, the trials included heterogeneous sample collection (retrospective plus prospective). NGS commercial panels have been included also in table 1, and TARDIS techniques have been described in the text. In breast metastatic cancer sections, hotspot mutations have been included in table 2.

Reviewer 2 Report

Comments to the manuscript ID cancers-1537340 are listed below:

  1. Although the abbreviation ctDNA is misunderstood, it is better to define the first time it is mentioned (line 19 page 1)
  2. In the text the figure is mentioned in general, clarify that it refers to figure 1, Indicate with dotted lines that item b) is an extension of item a) line 74 page 2, line 106 page 3
  3. Use the abbreviation ctDNA in lines 102, 104 page 3
  4. Page 7 line 300 homogenize the way of referencing sometimes the point is attached at the end and sometimes not (table 1).
  5. Table 2 write in italics when referring to genes

Author Response

Each suggestion provided have been addressed and changed.